# Co-Attentive Equivariant Neural Networks: Focusing Equivariance On Transformations Co-Occurring In Data

**David W. Romero**
Vrije Universiteit Amsterdam
`d.w.romeroguzman@vu.nl`

**Mark Hoogendoorn**
Vrije Universiteit Amsterdam
`m.hoogendoorn@vu.nl`

## Abstract

Equivariance is a nice property to have as it produces much more parameter efficient neural architectures and preserves the structure of the input through the feature mapping. Even though some combinations of transformations might never appear (e.g., an upright face with a horizontal nose), current equivariant architectures consider the set of all possible transformations in a transformation group when learning feature representations. Contrarily, the human visual system is able to attend to the set of relevant transformations occurring in the environment and utilizes this information to assist and improve object recognition. Based on this observation, we modify conventional equivariant feature mappings such that they are able to attend to the set of co-occurring transformations in data and generalize this notion to act on groups consisting of multiple symmetries. We show that our proposed *co-attentive equivariant neural networks* consistently outperform conventional rotation equivariant and rotation & reflection equivariant neural networks on rotated MNIST and CIFAR-10.

## 1 Introduction

Thorough experimentation in the fields of psychology and neuroscience has provided support to the intuition that our visual perception and cognition systems are able to identify familiar objects despite modifications in size, location, background, viewpoint and lighting (Bruce & Humphreys, 1994). Interestingly, we are not just able to recognize such modified objects, but are able to characterize which modifications have been applied to them as well. As an example, when we see a picture of a cat, we are not just able to tell that there is a cat in it, but also its position, its size, facts about the lighting conditions of the picture, and so forth. Such observations suggest that the human visual system is *equivariant* to a large *transformation group* containing translation, rotation, scaling, among others. In other words, the mental representation obtained by seeing a transformed version of an object, is equivalent to that of seeing the original object and transforming it mentally next.

These fascinating abilities exhibited by biological visual systems have inspired a large field of research towards the development of neural architectures able to replicate them. Among these, the most popular and successful approach is the Convolutional Neural Network (CNN) (LeCun et al., 1989), which incorporates equivariance to translation via convolution. Unfortunately, in counterpart to the human visual system, CNNs do not exhibit equivariance to other transformations encountered in visual data (e.g., rotations). Interestingly, however, if an ordinary CNN happens to learn rotated copies of the same filter, the stack of feature maps becomes equivariant to rotations even though individual feature maps are not (Cohen & Welling, 2016). Since ordinary CNNs must learn such rotated copies independently, they effectively utilize an important number of network parameters suboptimally to this end (see Fig. 3 in Krizhevsky et al. (2012)). Based on the idea that equivariance in CNNs can be extended to larger transformation groups by stacking convolutional feature maps, several approaches have emerged to extend equivariance to, e.g., planar rotations (Dieleman et al., 2016; Marcos et al., 2017; Weiler et al., 2018; Li et al., 2018), spherical rotations (Cohen et al., 2018; Worrall & Brostow, 2018; Cohen et al., 2019), scaling (Marcos et al., 2018; Worrall & Welling, 2019) and general transformation groups (Cohen & Welling, 2016), such that transformed copies of a single entity are not required to be learned independently.

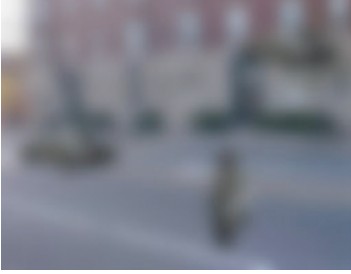

Figure 1: Our visual system infers object identities according to their size, location and orientation in a scene. In this blurred picture, observers describe the scene as containing a car and a pedestrian in the street. However, the pedestrian is in fact the same shape as the car, except for a 90° rotation. The atypicality of this orientation for a car *within the context defined by the street scene* causes the car to be recognized as a pedestrian. Extracted from Oliva & Torralba (2007).

Although incorporating equivariance to arbitrary transformation groups is conceptually and theoretically similar[1], evidence from real-world experiences motivating their integration might strongly differ. Several studies in neuroscience and psychology have shown that our visual system does not react equally to all transformations we encounter in visual data. Take, for instance, translation and rotation. Although we easily recognize objects independently of their position of appearance, a large corpus of experimental research has shown that this is not always the case for in-plane rotations. Yin (1969) showed that *mono-oriented objects*, i.e., complex objects such as faces which are customarily seen in one orientation, are much more difficult to be accurately recognized when presented upside-down. This behaviour has been reproduced, among others, for magazine covers (Dallett et al., 1968), symbols (Henle, 1942) and even familiar faces (e.g., from classmates) (Brooks & Goldstein, 1963). Intriguingly, Schwarzer (2000) found that this effect exacerbates with age (adults suffer from this effect much more than children), but, adults are much faster and accurate in detecting mono-oriented objects in usual orientations. Based on these studies, we draw the following conclusions:

- The human visual system does not perform (fully) equivariant feature transformations to visual data. Consequently, it does not react equally to all possible input transformations encountered in visual data, even if they belong to the same transformation group (e.g., in-plane rotations).

- The human visual system does not just encode familiarity to objects but seems to learn through experience the poses in which these objects customarily appear in the environment to assist and improve object recognition (Freire et al., 2000; Riesenhuber et al., 2004; Sinha et al., 2006).

Complementary studies (Tarr & Pinker, 1989; Oliva & Torralba, 2007) suggest that our visual system encodes orientation atypicality relative to the context rather than on an absolute manner (Fig. 1). Motivated by the aforementioned observations we state *the co-occurrence envelope hypothesis*:

**The Co-occurrence Envelope Hypothesis.** *By allowing equivariant feature mappings to detect transformations that co-occur in the data and focus learning on the set formed by these co-occurrent transformations (i.e., the co-occurrence envelope of the data), one is able to induce learning of more representative feature representations of the data, and, resultantly, enhance the descriptive power of neural networks utilizing them. We refer to one such feature mapping as* **co-attentive equivariant.**

**Identifying the co-occurrence envelope.** Consider a rotation equivariant network receiving two copies of the same face (Fig. 2a). A conventional rotation equivariant network is required to perform inference and learning on the set of all possible orientations of the visual patterns constituting a face regardless of the input orientation (Fig. 2b). However, by virtue of its rotation equivariance, it is able to recognize rotated faces even if it is trained on upright faces only. A possible strategy to simplify the task at hand could be to restrict the network to react exclusively to upright faces (Fig. 2c). In this case, the set of relevant visual pattern orientations becomes much smaller, at the expense of disrupting equivariance to the rotation group. Resultantly, the network would risk becoming unable to detect faces in any other orientation than those it is trained on. A better strategy results from restricting the set of relevant pattern orientations by defining them relative to one another

---

[1]It is achieved by developing feature mappings that utilize the transformation group in the feature mapping itself (e.g., translating a filter in the course of a feature transformation is used to obtain translation equivariance).

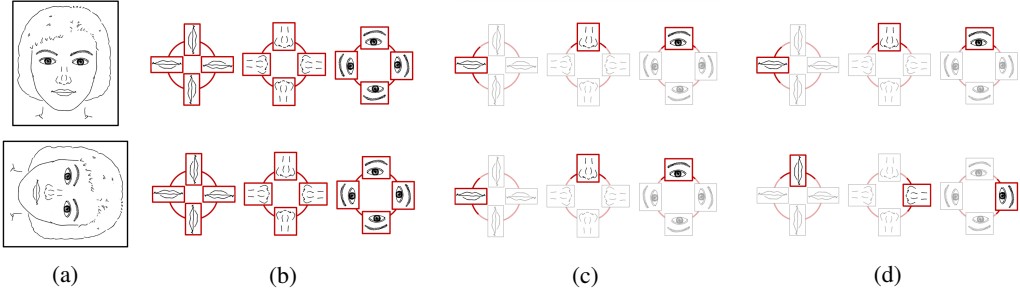

|     |     |     |     |
| :-: | :-: | :-: | :-: |
| (a) | (b) | (c) | (d) |

Figure 2: Effect of multiple attention strategies for the prioritization of relevant pattern orientations in rotation equivariant networks for the task of face recognition. Given that all attention strategies are learned exclusively from upright faces, we show the set of relevant directions for the recognition of faces in two orientations (Fig. 2a) obtained by: no attention (Fig. 2b), attending to the pattern orientations of appearance independently (Fig. 2c) and, attending to the pattern orientations of appearance relative to one another (Fig. 2d). Built upon Figure 1 from Schwarzer (2000).

(e.g., mouth orientation relative to the eyes) as opposed to absolutely (e.g., upright mouth) (Fig. 2d). In such a way, we are able to exploit information about orientation co-occurrences in the data without disrupting equivariance. The set of co-occurrent orientations in Fig. 2d corresponds to the co-occurrence envelope of the samples in Fig. 2a for the transformation group defined by rotations.

In this work, we introduce *co-attentive equivariant feature mappings* and apply them on existing equivariant neural architectures. To this end, we leverage the concept of *attention* (Bahdanau et al., 2014) to modify existing mathematical frameworks for equivariance, such that co-occurrent transformations can be detected. It is critical not to disrupt equivariance in the attention procedure as to preserve it across the entire network. To this end, we introduce *cyclic equivariant self-attention*, a novel attention mechanism able to preserve equivariance to cyclic groups.

**Experiments and results.** We explore the effects of co-attentive equivariant feature mappings for single and multiple symmetry groups. Specifically, we replace conventional rotation equivariant mappings in $p4$-CNNs (Cohen & Welling, 2016) and DRENs (Li et al., 2018) with co-attentive ones. We show that *co-attentive rotation equivariant neural networks* consistently outperform their conventional counterparts in fully (rotated MNIST) and partially (CIFAR-10) rotational settings. Subsequently, we generalize cyclic equivariant self-attention to multiple similarity groups and apply it on $p4m$-CNNs (Cohen & Welling, 2016) (equivariant to rotation and mirror reflections). Our results are in line with those obtained for single symmetry groups and support our stated hypothesis.

**Contributions.**

- We propose the *co-occurrence envelope hypothesis* and demonstrate that conventional equivariant mappings are consistently outperformed by our proposed *co-attentive equivariant* ones.

- We generalize co-attentive equivariant mappings to multiple symmetry groups and provide, to the best of our knowledge, the first attention mechanism acting generally on symmetry groups.

## 2 Preliminaries

**Equivariance.** We say that a feature mapping $f : X \rightarrow Y$ is equivariant to a (transformation) group $G$ (or $G$-equivariant) if it commutes with actions of the group $G$ acting on its domain and codomain:

$$f(T_g^X(x)) = T_g^Y(f(x)) \quad \forall g \in G, x \in X \tag{1}$$

where $T_g^{(\cdot)}$ denotes a *group action* in the corresponding space. In other words, the ordering in which we apply a group action $T_g$ and the feature mapping $f$ is inconsequential. There are multiple reasons as of why equivariant feature representations are advantageous for learning systems. Since group actions $T_g^X$ produce predictable and interpretable transformations $T_g^Y$ in the feature space, the *hypothesis space of the model* is reduced (Weiler et al., 2018) and the learning process simplified (Worrall et al., 2017). Moreover, equivariance allows the construction of $L$-layered networks by

stacking several equivariant feature mappings $\{f^{(1)}, ..., f^{(l)}, ..., f^{(L)}\}$ such that the input structure as regarded by the group $G$ is preserved (e.g., CNNs and input translations). As a result, an arbitrary intermediate network representation $(f^{(l)} \circ ... \circ f^{(1)})(x)$, $l \in L$, is able to take advantage of the structure of $x$ as well. *Invariance* is an special case of equivariance in which $T_g^Y = \mathrm{Id}_Y$, the identity, and thus all group actions in the input space are mapped to the same feature representation.

**Equivariant neural networks.** In neural networks, the integration of equivariance to arbitrary groups $G$ has been achieved by developing feature mappings $f$ that utilize the actions of the group $G$ in the feature mapping itself. Interestingly, *equivariant feature mappings* encode equivariance as *parameter sharing* with respect to $G$, i.e., the same weights are reused for all $g \in G$. This makes the inclusion of larger groups extremely appealing in the context of parameter efficient networks.

Conventionally, the $l$-th layer of a neural network receives a signal $x^{(l)}(u, \lambda)$ (where $u \in \mathbb{Z}^2$ is the spatial position and $\lambda \in \Lambda_l$ is the unstructured channel index, e.g., RGB channels in a color image), and applies a feature mapping $f^{(l)} : \mathbb{Z}^2 \times \Lambda_l \to \mathbb{Z}^2 \times \Lambda_{l+1}$ to generate the feature representation $x^{(l+1)}(u, \lambda)$. In CNNs, the feature mapping $f^{(l)} := f_T^{(l)}$ is defined by a *convolution*[2] ($\star_{\mathbb{R}^2}$) between the input signal $x^{(l)}$ and a learnable convolutional filter $W_{\lambda',\lambda}^{(l)}$, $\lambda' \in \Lambda_l$, $\lambda \in \Lambda_{l+1}$:

$$x^{(l+1)}(u, \lambda) = [x^{(l)} \star_{\mathbb{R}^2} W_{\lambda',\lambda}^{(l)}](u, \lambda) = \sum_{\lambda', u'} x^{(l)}(u + u', \lambda') W_{\lambda',\lambda}^{(l)}(u') \tag{2}$$

By sliding $W_{\lambda',\lambda}^{(l)}$ across $u$, CNNs are able to preserve the spatial structure of the input $x$ through the feature mapping $f_T^l$ and successfully provide equivariance to the translation group $T = (\mathbb{Z}^2, +)$.

The underlying idea for the extension of equivariance to larger groups in CNNs is conceptually equivalent to the strategy utilized by LeCun et al. (1989) for translation equivariance. Consider, for instance, the inclusion of equivariance to the set of rotations by $\theta_r$ degrees[3]: $\Theta = \{\theta_r = r\frac{2\pi}{r_{\max}}\}_{r=1}^{r_{\max}}$. To this end, we modify the feature mapping $f^{(l)} := f_R^{(l)} : \mathbb{Z}^2 \times \Theta \times \Lambda_l \to \mathbb{Z}^2 \times \Theta \times \Lambda_{l+1}$ to include the rotations defined by $\Theta$. Let $x^{(l)}(u, r, \lambda)$ and $W_{\lambda',\lambda}^{(l)}(u, r)$ be the input and the convolutional filter with an affixed index $r$ for rotation. The *roto-translational convolution* ($\star_{\mathbb{R}^2 \rtimes \Theta}$) $f_R^{(l)}$ is defined as:

$$x^{(l+1)}(u, r, \lambda) = [x^{(l)} \star_{\mathbb{R}^2 \rtimes \Theta} W_{\lambda',\lambda}^{(l)}](u, r, \lambda) = \sum_{\lambda', r', u'} x^{(l)}(u + u', r', \lambda') W_{\lambda',\lambda}^{(l)}(\theta_r u', r' - r) \tag{3}$$

Since $f_R^{(l)}$ produces ($\dim(\Theta) = r_{\max}$) times more output feature maps than $f_T^{(l)}$, we need to learn much smaller convolutional filters $W_{\lambda',\lambda}^{(l)}$ to produce the same number of output feature channels.

**Learning equivariant neural networks.** Consider the change of variables $g = u$, $G = \mathbb{Z}^2$, $g \in G$ and $g = (u, r)$, $G = \mathbb{Z}^2 \rtimes \Theta$, $g \in G$ in Eq. 2 and Eq. 3, respectively. In general, neural networks are learned via backpropagation (LeCun et al., 1989) by iteratively applying the chain rule of derivation to update the network parameters. Intuitively, the networks outlined in Eq. 2 and Eq. 3 obtain feedback from all $g \in G$ and, resultantly, are inclined to learn feature representations that perform optimally on the entire group $G$. However, as outlined in Fig. 2 and Section 1, several of those feature combinations are not likely to simultaneously appear. Resultantly, the *hypothesis space of the model* as defined by Weiler et al. (2018) might be further reduced.

Note that this reasoning is tightly related to existing explanations for the large success of *spatial* (Xu et al., 2015; Woo et al., 2018; Zhang et al., 2018) and *temporal* (Luong et al., 2015; Vaswani et al., 2017; Mishra et al., 2017; Zhang et al., 2018) attention in deep learning architectures.

## 3 CO-ATTENTIVE EQUIVARIANT NEURAL NETWORKS

In this section we define co-attentive feature mappings and apply them in the context of equivariant neural networks (Figure 3). To this end, we introduce cyclic equivariant self-attention and utilize it to construct co-attentive rotation equivariant neural networks. Subsequently, we show that cyclic equivariant self-attention is extendable to larger symmetry groups and make use of this fact to construct co-attentive neural networks equivariant to rotations and mirror reflections.

---

[2]Formally it is as a correlation. However, we hold on to the standard deep learning terminology.

[3]The reader may easily verify that $\Theta$ (and hence $\mathbb{Z}^2 \rtimes \Theta$, with ($\rtimes$) the semi-direct product) forms a group.

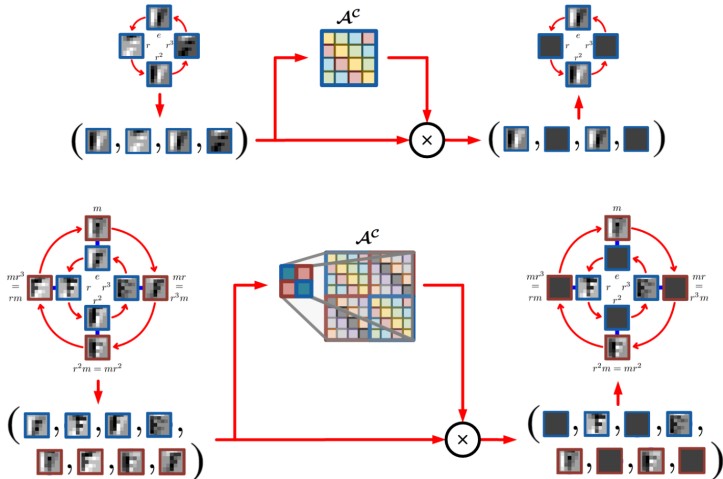

Figure 3: Co-attentive equivariant feature mappings acting on the groups $p4$ (top) and $p4m$ (bottom). In order to learn co-attentive equivariant representations, *cyclic equivariant self-attention* $\mathcal{A}^{\mathcal{C}}$ is applied on top of the output of a conventional equivariant feature mapping (here $p4$ and $p4m$ group convolutions, respectively). Resultantly, the group convolution responses are modulated based on their assessed relevance. For multiple symmetry groups, the group convolution responses must be rearranged in a vector structure so that the permutation laws of $\mathcal{A}^{\mathcal{C}}$ correspond to those of the composing group symmetries. Same colors in $\mathcal{A}^{\mathcal{C}}$ denote equal weights. The *circulant (block) structure* of $\mathcal{A}^{\mathcal{C}}$ ensures that equivariance to the corresponding group is preserved through the course of attention. Consequently, if the input is rotated (or mirrored in $p4m$), the attention mask shown here is transformed accordingly. Built upon Figures 1 and 2 from Cohen & Welling (2016).

## 3.1 CO-ATTENTIVE ROTATION EQUIVARIANT NEURAL NETWORKS

To allow rotation equivariant networks to utilize and learn *co-attentive equivariant representations*, we introduce an attention operator $\mathcal{A}^{(l)}$ on top of the roto-translational convolution $f_R^{(l)}$ with which discernment along the rotation axis $r$ of the generated feature responses $x^{(l)}(u, r, \lambda)$ is possible. Formally, our *co-attentive rotation equivariant feature mapping* $f_{\mathcal{R}}^{(l)}$ is defined as follows:

$$x^{(l+1)} = f_{\mathcal{R}}^{(l)}(x^{(l)}) = \mathcal{A}^{(l)}(f_R^{(l)}(x^{(l)})) = \mathcal{A}^{(l)}([x^{(l)} \star_{\mathbb{R}^2 \rtimes \Theta} W_{\lambda', \lambda}^{(l)}]) \qquad (4)$$

Theoretically, $\mathcal{A}^{(l)}$ could be defined globally over $f_R^{(l)}(x^{(l)})$ (i.e., simultaneously along $u$, $r$, $\lambda$) as depicted in Eq. 4. However, we apply attention locally to: (1) grant the algorithm enough flexibility to attend locally to the co-occurrence envelope of feature representations and, (2) utilize attention exclusively along the rotation axis $r$, such that our contributions are clearly separated from those possibly emerging from spatial attention. To this end, we apply attention pixel-wise on top of $f_R^{(l)}(x^{(l)})$ (Eq. 5). Furthermore, we assign a single attention instance $\mathcal{A}_\lambda^{(l)}$ to each learned feature representation and utilize it across the spatial dimension of the output feature maps[4]:

$$x^{(l+1)}(u, r, \lambda) = \mathcal{A}_\lambda^{(l)}(\{x^{(l+1)}(u, \hat{r}, \lambda)\}_{\hat{r}=1}^{r_{\max}})(r) \qquad (5)$$

**Attention and self-attention.** Consider a source vector $x = (x_1, ..., x_n)$ and a target vector $y = (y_1, ..., y_m)$. In general, an attention operator $\mathcal{A}$ leverages information from the source vector $x$ (or multiple feature mappings thereof) to estimate an attention matrix $A \in [0, 1]^{n \times m}$, such that: (1) the element $A_{i,j}$ provides an importance assessment of the source element $x_i$ with reference to the target element $y_j$ and (2) the sum of importance over all $x_i$ is equal to one: $\sum_i A_{i,j} = 1$. Subsequently, the matrix $A$ is utilized to modulate the original source vector $x$ as to *attend* to a subset of relevant source positions with regard to $y_j$: $\tilde{x}^j = (A_{:,j})^T \odot x$ (where $\odot$ is the Hadamard product). A special case of attention is that of *self-attention* (Cheng et al., 2016), in which the target and the source vectors are equal ($y := x$). In other words, the attention mechanism estimates the influence of the sequence $x$ on the element $x_j$ for its weighting.

---

[4]For a more meticulous discussion on how Eq. 5 attains co-occurrent attention, see Appendix A.

In general, the attention matrix[5] $A \in [0,1]^{n \times m}$ is constructed via nonlinear space transformations $f_{\tilde{A}} : \mathbb{R}^n \to \mathbb{R}^{n \times m}$ of the source vector $x$, on top of which the softmax function is applied: $A_{:,j} = \text{softmax}(f_{\tilde{A}}(x)_{:,j})$. This ensures that the properties previously mentioned hold. Typically, the mappings $f_{\tilde{A}}$ found in literature take feature transformation pairs of $x$ as input (e.g., $\{s, H\}$ in RNNs (Luong et al., 2015), $\{Q, K\}$ in self-attention networks (Vaswani et al., 2017)), and perform (non)-linear mappings on top of it, ranging from multiple feed-forward layers (Bahdanau et al., 2014) to several operations between the transformed pairs (Luong et al., 2015; Vaswani et al., 2017; Mishra et al., 2017; Zhang et al., 2018). Due to the computational complexity of these approaches and the fact that we do extensive pixel-wise usage of $f_{\tilde{A}}$ on every network layer, their direct integration in our framework is computationally prohibitive. To circumvent this problem, we modify the usual self-attention formulation as to enhance its descriptive power in a much more compact setting.

**Compact local self-attention.** Initially, we relax the range of values of $A$ from $[0,1]^{n \times n}$ to $\mathbb{R}^{n \times n}$. This allows us to encode much richer relationships between element pairs $(x_i, x_j)$ at the cost of less interpretability. Subsequently, we define $A = x^T \odot \tilde{A}$, where $\tilde{A} \in \mathbb{R}^{n \times n}$ is a matrix of learnable parameters. Furthermore, instead of directly applying softmax on the columns of $A$, we first sum over the contributions of each element $x_i$ to obtain a vector $a = \{\sum_i A_{i,j}\}_{j=1}^n$, which is then passed to the softmax function. Following Vaswani et al. (2017), we prevent the softmax function from reaching regions of low gradient by scaling its argument by $(\sqrt{\dim(A)})^{-1} = (1/n)$: $\tilde{a} = \text{softmax}((1/n)\, a)$. Lastly, we counteract the contractive behaviour of the softmax function by normalizing $\tilde{a}$ before weighting $x$ as to preserve the magnitude range of its argument. This allows us to use $\mathcal{A}$ in deep architectures. Our *compact self-attention mechanism* is summarized as follows:

$$a = \{\textstyle\sum_i A_{i,j}\}_{j=1}^n = \sum_i (x^T \odot \tilde{A})_{i,j} = x\tilde{A} \tag{6}$$

$$\tilde{a} = \text{softmax}((1/n)\, a) \tag{7}$$

$$\hat{x} = \mathcal{A}(x) = (\tilde{a} / \max(\tilde{a})) \odot x \tag{8}$$

**The cyclic equivariant self-attention operator $\mathcal{A}^{\mathcal{C}}$.** Consider $\{x(u, r, \lambda)\}_{r=1}^{r_{\max}}$, the vector of responses generated by a roto-translational convolution $f_R$ stacked along the rotation axis $r$. By applying self-attention along $r$, we are able to generate an importance matrix $A \in \mathbb{R}^{r_{\max} \times r_{\max}}$ relating all pairs of $(\theta_i, \theta_j)$-rotated responses in the rotational group $\Theta$ at a certain position. We refer to this attention mechanism as *full self-attention* ($\mathcal{A}^{\mathcal{F}}$). Although $\mathcal{A}^{\mathcal{F}}$ is able to encode arbitrary linear source-target relationships for each target position, it is not restricted to conserve equivariance to $\Theta$. Resultantly, we risk incurring into the behavior outlined in Fig. 2c. Before we further elaborate on this issue, we introduce the *cyclic permutation operator* $\mathcal{P}^i$, which induces a cyclic shift of $i$ positions on its argument: $\sigma^{\mathcal{P}^i}(x_j) = x_{(j+i) \bmod (\dim(x))}, \forall x_j \in x$.

Consider a full self-attention operator $\mathcal{A}^{\mathcal{F}}$ acting on top of a roto-translational convolution $f_R$. Let $p$ be an input pattern to which $f_R$ only produces a strong activation in the feature map $x(\hat{r}) = f_R(p)(\hat{r})$, $\hat{r} \in \{r\}_{r=1}^{r_{\max}}$. Intuitively, during learning, only the corresponding attention coefficients $\tilde{A}_{:,\hat{r}}$ in $\mathcal{A}^{\mathcal{F}}$ would be significantly increased. Now, consider the presence of the input pattern $\theta_i p$, a $\theta_i$-rotated variant of $p$. By virtue of the rotational equivariance property of the feature mapping $f_R$, we obtain (locally) an exactly equal response to that of $p$ up to a cyclic permutation of $i$ positions on $r$, and thus, we obtain a strong activation in the feature map $\mathcal{P}^i(x(\hat{r})) = x(\sigma^{\mathcal{P}^i}(\hat{r}))$. We encounter two problems in this setting: $\mathcal{A}^{\mathcal{F}}$ is not be able to detect that $p$ and $\theta_i p$ correspond to the exact same input pattern and, as each but the attention coefficients $\tilde{A}_{:,j}$ is small, the network might considerably damp the response generated by $\theta_i p$. As a result, the network might (1) squander important feedback information during learning and (2) induce learning of repeated versions of the same pattern for different orientations. In other words, $\mathcal{A}^{\mathcal{F}}$ does not behave equivariantly as a function of $\theta_i$.

Interestingly, we are able to introduce prior-knowledge into the attention model by restricting the structure of $\tilde{A}$. By leveraging the idea of *equivariance to the cyclic group* $\mathcal{C}_n$, we are able to solve the problems exhibited by $\mathcal{A}^{\mathcal{F}}$ and simultaneously reduce the number of additional parameters required by the self-attention mechanism (from $r_{\max}^2$ to $r_{\max}$). Consider again the input patterns $p$ and $\theta_i p$. We incorporate the intuition that $p$ and $\theta_i p$ are one and the same entity, and thus, $f_R$ (locally) generates the same output feature map up to a cyclic permutation $\mathcal{P}^i$: $f_R(\theta_i p) = \mathcal{P}^i(f_R(p))$. Consequently, the attention mechanism should produce the *exact same* output for both $p$ and $\theta_i p$ up to the same cyclic permutation $\mathcal{P}^i$. In other words, $\mathcal{A}$ (and thus $\tilde{A}$) should be *equivariant to cyclic permutations*.

---

[5]Technically, each column of $A$ is restricted to a simplex and hence $A$ lives in a subspace of $[0,1]^{n \times m}$.

A well-known fact in mathematics is that a matrix $A$ is equivariant with respect to cyclic permutations of the domain if *and only if* it is *circulant* (Alkarni, 2001; Åhlander & Munthe-Kaas, 2005). We make use of this certitude and leverage the concept of *circulant matrices* to impose cyclic equivariance to the structure of $\tilde{A}$. Formally, a circulant matrix $C \in \mathbb{R}^{n \times n}$ is composed of $n$ cyclic permutations of its defining vector $c = \{c_i\}_{i=1}^n$, such that its $j$-th column is a cyclic permutation of $j - 1$ positions of $c$: $C_{:,j} = \mathcal{P}^{j-1}(c)^T$. We construct our *cyclic equivariant self-attention operator* $\mathcal{A}^{\mathcal{C}}$ by defining $\tilde{A}$ as a circulant matrix specified by a learnable attention vector $a^{\mathcal{C}} = \{a_i^{\mathcal{C}}\}_{i=1}^{r_{\max}}$:

$$\tilde{A} = \{\mathcal{P}^{j-1}(a^{\mathcal{C}})^T\}_{j=1}^n \tag{9}$$

and subsequently applying Eqs. 6 - 8. Resultantly, $\mathcal{A}^{\mathcal{C}}$ is able to assign the responses generated by $f_R$ for rotated versions of an input pattern $p$ to a unique entity: $f_R(\theta_i p) = \mathcal{P}^i(f_R(p))$, and dynamically adjust its output to the angle of appearance $\theta_i$, such that the attention operation does not disrupt its propagation downstream the network: $\mathcal{A}^{\mathcal{C}}(f_R(\theta_i p)) = \mathcal{P}^i(\mathcal{A}^{\mathcal{C}}(f_R(p)))$. Consequently, the attention weights $a^{\mathcal{C}}$ are updated equally regardless of specific values of $\theta_i$. Due to these properties, $\mathcal{A}^{\mathcal{C}}$ does not incur in any of the problems outlined earlier in this section. Conclusively, our *co-attentive rotation equivariant feature mapping* $f_{\mathcal{R}}^{(l)}$ is defined as follows:

$$x^{(l+1)}(u, r, \lambda) = f_{\mathcal{R}}^{(l)}(x^{(l)})(u, r, \lambda) = \mathcal{A}_{\lambda}^{\mathcal{C}(l)}\big([x^{(l)} \star_{\mathbb{R}^2 \rtimes \Theta} W_{\lambda', \lambda}^{(l)}]\big)(u, r, \lambda) \tag{10}$$

Note that a co-attentive equivariant feature mapping $f_{\mathcal{R}}$ is approximately equal (up to a normalized softmax operation (Eq. 8)) to a conventional equivariant one $f_R$, if $\tilde{A} = \alpha I$ for any $\alpha \in \mathbb{R}$.

## 3.2 Extending $\mathcal{A}^{\mathcal{C}}$ to Multiple Symmetry Groups

The self-attention mechanisms outlined in the previous section are easily extendable to larger groups consisting of multiple symmetries. Consider, for instance, the group $\theta_r m$ of rotations by $\theta_r$ degrees and mirror reflections $m$ defined analogously to the group $p4m$ in Cohen & Welling (2016). Let $p(u, r, m, \lambda)$ be an input signal with an affixed index $m \in \{m_0, m_1\}$ for mirror reflections ($m_1$ indicates mirrored) and $f_{\theta_r m}$ be a *group convolution* (Cohen & Welling, 2016) on the $\theta_r m$ group. The group convolution $f_{\theta_r m}$ produces two times as many output channels ($2r_{\max} : m_0 r_{\max} + m_1 r_{\max}$) as those generated by the roto-translational convolution $f_R$ (Eq. 3, Fig. 3).

Full self-attention $\mathcal{A}^{\mathcal{F}}$ can be integrated directly by modulating the output of $f_{\theta_r m}$ as depicted in Sec. 3.1 with $\tilde{A} \in \mathbb{R}^{2r_{\max} \times 2r_{\max}}$. Here, $\mathcal{A}^{\mathcal{F}}$ relates each of the group convolution responses with one another. However, just as for $f_R$, $\mathcal{A}^{\mathcal{F}}$ disrupts the equivariance property of $f_{\theta_r m}$ to the $\theta_r m$ group.

Similarly, the cyclic equivariant self-attention operator $\mathcal{A}^{\mathcal{C}}$ can be extended to multiple symmetry groups as well. Before we continue, we introduce the *cyclic permutation operator* $\mathcal{P}^{i,t}$, which induces a cyclic shift of $i$ positions on its argument along the transformation axis $t$. Consider the input patterns $p$ and $\theta_i p$ outlined in the previous section and $mp$, a mirrored instance of $p$. Let $x(u, r, m, \lambda) = f_{\theta_r m}(p)(u, r, m, \lambda)$ be the response of the group convolution $f_{\theta_r m}$ for the input pattern $p$. By virtue of the rotation equivariance property of $f_{\theta_r m}$, the generated response for $\theta_i p$ is equivalent to that of $p$ up to a cyclic permutation of $i$ positions along the rotation axis $r$: $f_{\theta_r m}(\theta_i p)(u, r, m, \lambda) = \mathcal{P}^{i,r}(f_{\theta_r m}(p))(u, r, m, \lambda) = x(u, \sigma^{\mathcal{P}^i}(r), m, \lambda)$. Similarly, by virtue of the mirror equivariance property of $f_{\theta_r m}$, the response generated by $mp$ is equivalent to that of $p$ up to a cyclic permutation of one position along the mirroring axis $m$: $f_{\theta_r m}(mp)(u, r, m, \lambda) = \mathcal{P}^{1,m}(f_{\theta_r m}(p))(u, r, m, \lambda) = x(u, r, \sigma^{\mathcal{P}^1}(m), \lambda)$. Note that if we take two elements from a group $g$, $h$, their composition $(gh)$ is also an element of the group. Resultantly, $f_{\theta_r m}((m\theta^i)p)(u, r, m, \lambda) = (\mathcal{P}^{1,m} \circ \mathcal{P}^{i,r})(f_{\theta_r m}(p))(u, r, m, \lambda) = \mathcal{P}^{1,m}(\mathcal{P}^{i,r}(x))(u, r, m, \lambda) = \mathcal{P}^{1,m}(x)(u, \sigma^{\mathcal{P}^i}(r), m, \lambda) = x(u, \sigma^{\mathcal{P}^i}(r), \sigma^{\mathcal{P}^1}(m), \lambda)$.

In other words, in order to extend $\mathcal{A}^{\mathcal{C}}$ to the $\theta_r m$ group, it is necessary to restrict the structure of $\tilde{A}$ such that it respects the *permutation laws* imposed by the equivariant mapping $f_{\theta_r m}$. Let us rewrite $x(u, r, m, \lambda)$ as $x(u, g, \lambda)$, $g = (mr) \in \{m_0, m_1\} \times \{\hat{r}\}_{\hat{r}=1}^{r_{\max}}$. In this case, we must impose a *circulant block matrix* structure on $\tilde{A}$ such that: (1) the composing blocks permute internally as defined by $\mathcal{P}^{i,r}$ and (2) the blocks themselves permute with one another as defined by $\mathcal{P}^{1,m}$. Formally, $\tilde{A}$ is defined as:

$$\tilde{A} = \begin{bmatrix} \tilde{A}_1 & \tilde{A}_2 \\ \tilde{A}_2 & \tilde{A}_1 \end{bmatrix} \tag{11}$$

where $\{\tilde{A}_i \in \mathbb{R}^{r_{\max} \times r_{\max}}\}$, $i \in \{1, 2\}$ are circulant matrices (Eq. 9). Importantly, the ordering of the permutation laws in $\tilde{A}$ is interchangeable if the input vector is modified accordingly, i.e., $g = (rm)$.

Conclusively, cyclic equivariant self-attention $\mathcal{A}^\mathcal{C}$ is directly extendable to act on any $G$-equivariant feature mapping $f_G$, and for any symmetry group $G$, if the group actions $T_g^Y$ produce cyclic permutations on the codomain of $f_G$. To this end, one must restrict the structure of $\tilde{A}$ to that of a circulant block matrix, such that *all* permutation laws of $T_g^Y$ hold: $T_g^Y(\mathcal{A}^\mathcal{C}(f_G)) = \mathcal{A}^\mathcal{C}(T_g^Y(f_G)), \forall g \in G$.

## 4 EXPERIMENTS

**Experimental Setup.** We validate our approach by exploring the effects of co-attentive equivariant feature mappings for single and multiple symmetry groups on existing equivariant architectures. Specifically, we replace conventional rotation equivariant mappings in $p4$-CNNs (Cohen & Welling, 2016) and DRENs (Li et al., 2018) with co-attentive equivariant ones and evaluate their effects in fully (rotated MNIST) and partially (CIFAR-10) rotational settings. Similarly, we evaluate co-attentive equivariant maps acting on multiple similarity groups by replacing equivariant mappings in $p4m$-CNNs (Cohen & Welling, 2016) (equivariant to rotation and mirror reflections) likewise. Unless otherwise specified, we replicate as close as possible the same data processing, initialization strategies, hyperparameter values and evaluation strategies utilized by the baselines in our experiments. Note that the goal of this paper is to study and evaluate the relative effects obtained by co-attentive equivariant networks with regard to their conventional counterparts. Accordingly, we do not perform any additional tuning relative to the baselines. We believe that improvements on our reported results are feasible by performing further parameter tuning (e.g., on the network structure or the used hyperparameters) on the proposed co-attentive equivariant networks.

The additional learnable parameters, i.e., those associated to the cyclic self-attention operator ($\tilde{A}$) are initialized identically to the rest of the layer. Subsequently, we replace the values of $\tilde{A}$ along the diagonal by 1 (i.e., $\text{diag}(\tilde{A}_{\text{init}}) = 1$) such that $\tilde{A}_{\text{init}}$ approximately resembles the identity $I$ and, hence, co-attentive equivariant layers are initially approximately equal to equivariant ones.

**Rotated MNIST.** The rotated MNIST dataset (Larochelle et al., 2007) contains 62000 gray-scale 28x28 handwritten digits uniformly rotated on the entire circle $[0, 2\pi)$. The dataset is split into training, validation and tests sets of 10000, 2000 and 50000 samples, respectively. We replace rotation equivariant layers in $p4$-CNN (Cohen & Welling, 2016), DREN and DRENMaxPooling (Li et al., 2018) with co-attentive ones. Our results show that co-attentive equivariant networks consistently outperform conventional ones (see Table 1).

**CIFAR-10.** The CIFAR-10 dataset (Krizhevsky et al., 2009) consists of 60000 real-world 32x32 RGB images uniformly drawn from 10 classes. Contrarily to the rotated MNIST dataset, this dataset does not exhibit rotation symmetry. The dataset is split into training, validation and tests sets of 40000, 10000 and 10000 samples, respectively. We replace equivariant layers in the $p4$ and $p4m$ variations of the All-CNN (Springenberg et al., 2014) and the ResNet44 (He et al., 2016) proposed by Cohen & Welling (2016) with co-attentive ones. Likewise, we modify the r_x4-variations of the NIN (Lin et al., 2013) and ResNet20 (He et al., 2016) models proposed by Li et al. (2018) in the same manner. Our results show that co-attentive equivariant networks consistently outperform conventional ones in this setting as well (see Table 1).

**Training convergence of equivariant networks.** Li et al. (2018) reported that adding too many rotational equivariant (isotonic) layers decreased the performance of their models on CIFAR-10. As a consequence, they did not report results on fully rotational equivariant networks for this setting and attributed this behaviour to the non-symmetricity of the data. We noticed that, with equal initialization strategies, rotational equivariant CNNs were much more prone to divergence than ordinary CNNs. This behaviour can be traced back to the additional feedback resulting from roto-translational convolutions (Eq. 3) compared to ordinary ones (Eq. 2). After further analysis, we noticed that the data preprocessing strategy utilized by Li et al. (2018) leaves some very large outlier values in the data ($|x| > 100$), which strongly contribute to the behaviour outlined before.

In order to evaluate the relative contribution of co-attentive equivariant neural networks we constructed fully equivariant DREN architectures based on their implementation. Although the obtained results were much worse than those originally reported in Li et al. (2018), we were able to stabilize

Table 1: Comparison of conventional equivariant and co-attentive equivariant neural networks. Values between parenthesis correspond to relevant results obtained from our own experiments.

| Rotated MNIST | | | CIFAR-10 | | |
|---|---|---|---|---|---|
| Network | Test Error (%) | Param. | Network | Test Error (%) | Param. |
| Z2CNN | $5.03 \pm 0.002$ | 21.75k | All-CNN | 9.44 | 1.372M |
| $p4$-CNN | $2.28 \pm 0.0004$ | 19.88k | $p4$-All-CNN | 8.84 | 1.371M |
| $a$-$p4$-CNN | $\mathbf{2.06 \pm 0.0429}$ | 20.76k | $a$-$p4$-All-CNN | **7.68** | 1.373M |
| DREN | 1.78 (1.99) | 19.88k | $p4m$-All-CNN | 7.59 | 1.219M |
| $a$-DREN | **1.674** | 20.76k | $a$-$p4m$-All-CNN | **6.92** | 1.223M |
| DRENMaxPool. | 1.56 (1.60) | 24.68k | ResNet44 | 9.45 (9.85) | 2.639M |
| $a$-DRENMaxPool. | **1.34** | 25.68k | $p4m$-ResNet44 | 6.46 (9.47) | 2.623M |
| | | | $a$-$p4m$-ResNet44 | **9.12** | 2.632M |
| | | | NIN | 10.41 (15.92) | 0.967M |
| | | | r-NINx4 | 14.96 | 0.958M |
| | | | $a$-r-NINx4 | **13.67** | 0.968M |
| | | | ResNet20 | 9.00 (12.32) | 0.335M |
| | | | r-ResNet20x4 | 12.31 | 0.333M |
| | | | $a$-r-ResNet20x4 | **11.32** | 0.339M |

training by clipping input values to the 99 percentile of the data ($|x| \leq 2.3$) and reducing the learning rate to 0.01, such that the same hyperparameters could be used across all network types. The obtained results (see Table 1) signalize that DREN networks are *comparatively better than CNNs both in fully and partially rotational settings*, contradictorily to the conclusions drawn in Li et al. (2018).

This behaviour elucidates that although the inclusion of equivariance to larger transformation groups is beneficial both in terms of accuracy and parameter efficiency, one must be aware that such benefits are directly associated with an increase of the network susceptibility to divergence during training.

## 5 DISCUSSION AND FUTURE WORK

Our results show that co-attentive equivariant feature mappings can be utilized to enhance conventional equivariant ones. Interestingly, co-attentive equivariant mappings are beneficial both in partially and fully rotational settings. We attribute this to the fact that a set of co-occurring orientations between patterns can be easily defined (and exploited) in both settings. It is important to note that we utilized attention independently over each spatial position $u$ on the codomain of the corresponding group convolution. Resultantly, we were restricted to mappings of the form $xA$, which, in turn, constraint our attention mechanism to have a circulant structure in order to preserve equivariance (since group actions acting in the codomain of the group convolution involve cyclic permutations and cyclic self-attention is applied in the codomain of the group convolution).

In future work, we want to extend the idea presented here to act on the entire group simultaneously (i.e., along $u$ as well). By doing so, we lift our current restriction to mappings of the form $xA$ and therefore, may be able to develop attention instances with enhanced descriptive power. Following the same line of though, we want to explore incorporating attention in the convolution operation itself. Resultantly, one is not restricted to act exclusively on the codomain of the convolution, but instead, is able to impose structure in the domain of the mapping as well. Naturally, such an approach could lead to enhanced descriptiveness of the incorporated attention mechanism. Moreover, we want to utilize and extend more complex attention strategies (e.g., Bahdanau et al. (2014); Luong et al. (2015); Vaswani et al. (2017); Mishra et al. (2017)) such that they can be applied to large transformation groups without disrupting equivariance. As outlined earlier in Section 3.1, this becomes very challenging from a computational perspective as well, as it requires extensive usage of the corresponding attention mechanism. Resultantly, an efficient implementation thereof is mandatory. Furthermore, we want to extend co-attentive equivariant feature mappings to continuous (e.g., Worrall et al. (2017)) and 3D space (e.g., Cohen et al. (2018); Worrall & Brostow (2018); Cohen et al. (2019)) groups, and for applications other than visual data (e.g., speech recognition).

Finally, we believe that our approach could be refined and extended to a first step towards dealing with the enumeration problem of large groups (Gens & Domingos, 2014), such that functions acting on the group (e.g., group convolution) are approximated by evaluating them on the set of co-occurring transformations as opposed to on the entire group. Such approximations are expected to be very accurate, as non-co-occurrent transformations are rare. This could be though of as sharping up co-occurrent attention to co-occurrent restriction.

## 6 CONCLUSION

We have introduced the concept of co-attentive equivariant feature mapping and applied it in the context of equivariant neural networks. By attending to the co-occurrence envelope of the data, we are able to improve the performance of conventional equivariant ones on fully (rotated MNIST) and partially (CIFAR-10) rotational settings. We developed cyclic equivariant self-attention, an attention mechanism able to attend to the co-occurrence envelope of the data without disrupting equivariance to a large set of transformation groups (i.e., all transformation groups $G$, whose action in the codomain of a $G$-equivariant feature mapping produce cyclic permutations). Our obtained results support the proposed co-occurrence envelope hypothesis.

### ACKNOWLEDGMENTS

We gratefully acknowledge Jan Klein, Emile van Krieken, Jakub Tomczak and our anonymous reviewers for their helpful and valuable commentaries. This work is part of the Efficient Deep Learning (EDL) programme (grant number P16-25), which is partly founded by the Dutch Research Council (NWO) and Semiotic Labs.

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

# A  OBTAINING CO-OCCURRENT ATTENTION VIA EQUATION 5

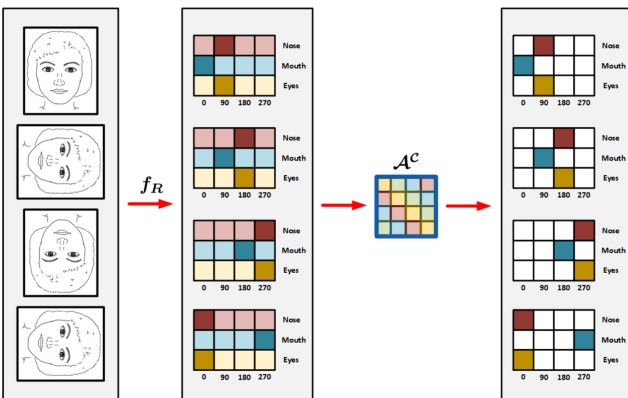

Figure 4: Synchronous movement of feature mappings and attention masks as a function of input rotation in the group $p4$ ($r_{\max} = 4$).

In this section, we provide a meticulous description on how co-occurrent attention is obtained via the method presented in the paper. Intuitively, a direct approach to address the problem illustrated in the introduction (Section 1) and Figure 2 requires an attention mechanism that acts simultaneously on $r$ and $\lambda$ (see Eq. 3). However, we illustrate how the depicted problem can be simplified such that attention along $r$ is sufficient by taking advantage of the equivariance property of the network.

Let $p$ be the input of a roto-translational convolution $f_R : \mathbb{Z}^2 \times \Theta \times \Lambda_0 \rightarrow \mathbb{Z}^2 \times \Theta \times \Lambda_1$ as defined in Eq. 3, and $\Theta$ be the set of rotations by $\theta_r$ degrees: $\Theta = \{\theta_r = r\frac{2\pi}{r_{\max}}\}_{r=1}^{r_{\max}}$. Let $f_R(p)(u) \in \mathbb{R}^{r_{\max} \times \Lambda_1}$ be the matrix consisting of the $r_{\max}$ oriented responses for each $\lambda \in \Lambda_1$ learned representation at a certain position $u$. Since the vectors $f_R(p)(u, \lambda) \in \mathbb{R}^{r_{\max}}$, $\lambda \in \Lambda_1$ permute cyclically as a result of the rotation equvivariance property of $f_R$, it is mandatory to ensure equivariance to cyclic permutations for each $f_R(p)(u, \lambda)$ during the course of the attention procedure (see Section 3).

At first sight, one is inclined to think that there is no connection between multiple vectors $f_R(p)(u, \lambda)$ in $f_R(p)(u)$, and, therefore, in order to exploit co-occurences, one must impose additional constraints along the $\lambda$ axis. However, there is indeed an implicit restriction in $f_R(p)(u)$ along $\lambda$ resulting from the rotation equivariance property of the mapping $f_R$, which we can take advantage from to simplify the problem at hand. Consider, for instance, the input $\theta_i p$, a $\theta_i$-rotated version of $p$. By virtue of the equivariance property of $f_R$, we have (locally) that $f_R(\theta_i p) = \mathcal{P}^i(f_R(p))$. Furthermore, we know that this property must hold for all the learned feature representations $f_R(p)(u, \lambda), \forall \lambda \in \Lambda_1$. Resultantly, we have that:

$$f_R(\theta_i p)(u, r, \lambda) = \mathcal{P}^i(f_R(p)(u, r, \lambda)) \ , \ \ \forall \lambda \in \Lambda_1 \tag{12}$$

In other words, if one of the learned mappings $f_R(p)(u, r, \lambda)$ experiences a permutation $\mathcal{P}^i$ along $r$, *all* the learned representations $f_R(p)(u, r, \lambda), \forall \lambda \in \Lambda_1$ must experience the exact same permutation $\mathcal{P}^i$ as well. Resultantly, the equivariance property of the mapping $f_R$ ensures that *all* the $\Lambda_1$ learned feature representations $f_R(p)(u, \lambda)$ *"move synchronously"* as a function of input rotation $\theta_i$.

Likewise, if we apply a cyclic equivariant attention mechanism $\mathcal{A}_\lambda^{\mathcal{C}}$ independently on top of each $\lambda$ learned representation $f_R(p)(u, \lambda)$, we obtain that the relation

$$\mathcal{A}_\lambda^{\mathcal{C}}(f_R(\theta_i p))(u, r, \lambda) = \mathcal{P}^i(\mathcal{A}_\lambda^{\mathcal{C}}(f_R(p))(u, r, \lambda)) \ , \ \ \forall \lambda \in \Lambda_1 \tag{13}$$

must hold as well. Similarly to the case illustrated in Eq. 12 and given that $\mathcal{A}_\lambda^{\mathcal{C}}$ is equivariant to cyclic permutations on the domain, we obtain that *all* the $\Lambda_1$ learned *attention masks* $\mathcal{A}_\lambda^{\mathcal{C}}$ *"move synchronously"* as a function of input rotation $\theta_i$ as well (see Fig. 4).

From Eq. 13 and Figure 4, one can clearly see that by utilizing $\mathcal{A}_\lambda^{\mathcal{C}}$ independently along $r$ and taking advantage from the fact that all $\Lambda_1$ learned feature representations are tied with one another via $f_R$, one is able to prioritize learning of feature representations that co-occur together as opposed to the much looser formulation in Eq. 12, where feedback is obtained from all orientations.

