# OpenReview forum: "Co-Attentive Equivariant Neural Networks: Focusing Equivariance On Transformations Co-Occurring in Data"
_ICLR.cc/2020/Conference — Accept (Poster)_

### Official Review · AnonReviewer3 · 2019-10-19
**Official Blind Review #3**

**Rating:** 6

**Review:**

[Post-rebuttal update]

Having read the rebuttals and seen the new draft, the authors have answered a lot of my concerns. I am still unsatisfied about the experimental contribution, but I guess producing a paper full of theory and good experiments is a tall ask. Having also read through the concerns of the other reviews and the rebuttal to them, I have decided to upgrade my review to a 6.

*Paper summary*

The paper combines attention with group equivariance, specifically looking at the p4m group of rotations, translations, and flips. The basic premise is to use a group equivariant CNN of, say, Cohen and Welling (2016), and use self-attention on top. The authors derive a form of self-attention that does not destroy the equivariance property.


*Paper decision*

I have decided that the paper be given a weak reject. The method seems sound and I think this in itself is a great achievement., But the experiments lack focus. Just showing that you get better accuracy results does not actually test why attention helps in an equivariant setting. That said, I feel the lack of clarity in the writing is actually the main drawback. The maths is poorly explained and the technical jargon is quite confusing. I think this can be improved in a camera-ready version or in submission to a later conference, should overall acceptance not be met.


*Supporting arguments*

I enjoyed the motivation and discussion on equivariance from a neuroscientific perspective. This is something I have not seen much of in the recent literature (which is more mathematical in nature) and serves as a refreshing take on the matter. There was a good review of the neuroscientific literature and I felt that the conclusions, which were draw (of approximate equivariance, and learned canonical transformations) were well motivated by these paper.

The paper is well structured. That said, I found the clarity of the technical language at times quite difficult to follow because terms were not defined. By way of example, I still have trouble understanding terms like “co-occurence” or “dynamically learn”. In the co-occurence envelope hypothesis, for instance, what does it mean for a learned feature representation to be “optimal in the set of transformations that co-occur”. Against what metric exactly would a representation be optimal? This is not defined.

That said, I feel that the content and conclusions of the paper are technically sound, having followed the maths, because the text was too confusing.


*Questions/notes for the authors*

- I would like to know whether the co-occurence envelope hypothesis is the authors’ own contribution. This was not apparent to me from the text.
- I’m not sure what exactly the co-occurence envelope is. It does not seem to be defined very precisely. What is it in layman’s terms?
- I found the section “Identifying the co-occurence envelope” very confusing. I’m not sure what the authors are trying to explain here. Is it that a good feature representation of a face would use the *relevant* offsets/rotations/etc. of visual features from different parts of the face, independent of global rotation?
- Is Figure 1 supposed to be blurry?
- At the end of paragraph 1 you have written: sdfgsdfg asdfasdf. Please delete this.
- I believe equation 4 is a roto-translational convolution since it is equivariant to rotation *and translation*. Furthermore, it is not exactly equivariant due to the fact that you are defining input on a 2D square grid, but that is a minor detail in the context of this work.
- Now that we have automatic differentiation, is the section on how to work out the gradients in Equations 5-7 really necessary?
- In equation 8 (second equality), you have said f_R^l(F^l) = A(f_R^l(F^l)). How can this be true if A is not the identity? Giving the benefit of the doubt, this could just be a typo.
- Please define \odot (I think it’s element-wise multiplication).
- Are you using row-vector convention? That would resolve some of my confusion with the maths.
- You define the matrix A as in the space [0,1]^{n x m}. While sort of true, it is more precise to note that each column is actually restricted to a simplex, so A lives in a subspace of [0,1]^{n x m}.
- I think it would have been easier just say that you are using a roto-translation or p4m equivariant CNN with attention after each convolution. Then you could derive the constraint on the attention matrices to maintain equivariance. It would be easier to follow and make easy connections with existing literature on the topic.

**Experience Assessment:**

I have published in this field for several years.

**Review Assessment: Checking Correctness Of Derivations And Theory:**

I carefully checked the derivations and theory.

**Review Assessment: Checking Correctness Of Experiments:**

I assessed the sensibility of the experiments.

**Review Assessment: Thoroughness In Paper Reading:**

I read the paper at least twice and used my best judgement in assessing the paper.

---

> ### Author Response · Authors · 2019-11-06
> **First thoughts**
>
> Dear reviewer 3,
>
> First of all thank you very much for thorough review and, of course, for your time.
>
> 1. We consistently obtained from all the reviewers the observation that the notation and technical jargon utilized to explain our approach was excessive. We agree with this.
>
> Our actual motivation to utilize the current notation comes from the  *Default Notation* section of the conference template's ( https://github.com/ICLR/Master-Template/blob/master/archive/iclr2020.zip ). Here, it is encouraged to use the notation utilized in Godfellow et. al. (2016), which (to some extent) contributes to the excessive technical jargon. Several of the equations appearing in our *Section 2* are indeed slight modifications of equations appearing in the *Convolutional Networks* chapter of Godfellow et. al. (2016), which we subsequently utilize to define our own approach.
>
> Furthermore, we accept that we did not introduce relevant terms properly. This is also related to the fact that we understood the provided notation as (intended to be) standard for all submissions and, hence, we did not feel obliged to introduce notation defined in there (e.g. row-vector convention). Naturally, this also contributes to the non-clarity of the derivations.
>
> That said, we accept that the aforementioned facts are not the only reasons contributing to non-clarity. We will work hard at making them much clearer in a subsequent version of our work. We will utilize a simpler, well-introduced notation as well.
>
> 2. Regarding your arguments:
> - We will be careful to introduce all terms utilized in the text. e.g. "co-occurrence".
> - We will add further explanations as to what is to be "optimal in the set of transformations that co-occur".
>
> 3. Regarding your questions:
>
> Q: I would like to know whether the co-occurence envelope hypothesis is the authors’ own contribution. This was not apparent to me from the text.
> A: Yes it is (it will be made clear in the following paper version)
>
> Q:  I’m not sure what exactly the co-occurence envelope is. It does not seem to be defined very precisely. What is it in layman’s terms?
> A: The co-occurrence envelope refers to the set of actual co-occurring transformations in a group. Consider, for example, Figure 2. Assume we have eye's, mouth's and nose's detectors. When detecting faces, a conventional equiv. net would reason in the space formed by all possible orientations of each feature (in this case eye, mouth, nose) (Fig. 2b). If one were able to define which feature orientations actually appear together, one would be able to ignore feature poses that do not (e.g. a face cannot be constructed with an upright nose and a vertical mouth). The set of feature orientations that appear together (co-occur), compose the set of co-occurring transformations or, in other words, the co-occurrence envelope of that face (See Fig 2.d).
>
> Q: - I found the section “Identifying the co-occurence envelope” very confusing. I’m not sure what the authors are trying to explain here. Is it that a good feature representation of a face would use the *relevant* offsets/rotations/etc. of visual features from different parts of the face, independent of global rotation?
> A: Precisely. Although we do not explore offsets in our work, that is indeed the intuition behind it. Maybe our answer to the previous question can shed some light here too. We will make use of your observations to construct a clearer definition.
>
> Q: Is Figure 1 supposed to be blurry?
> A: Yes. The purpose of the image is to show that humans intuitively connect relative orientations of objects to make assumptions about them. Note that one naturally assumes the horizontal blurred shape to be a car and the exact same shape rotated to be a person (although it is indeed the same blurred car). This is due to the atypicality of a vertical orientation for a car within the context defined by the scene.
>
> Q: At the end of paragraph 1 you have written: sdfgsdfg asdfasdf. Please delete this.
> A: Sorry about that.
>
> Q: I believe equation 4 is a roto-translational convolution since it is equivariant to rotation *and translation*. Furthermore, it is not exactly equivariant due to the fact that you are defining input on a 2D square grid, but that is a minor detail in the context of this work.
> A: You are right. It will be corrected.
>
> Q: Now that we have automatic differentiation, is the section on how to work out the gradients in Equations 5-7 really necessary?
> A: It is intended to show that parameters receive more information as more weight-tying is utilized. Additionally, it sheds light on the behavior of DREN networks (Li et al., 2018) in the section *Training convergence of equivariant networks*. We will consider perhaps moving this section to the appendix.
>
> [1] Ian Goodfellow, Yoshua Bengio, Aaron Courville, and Yoshua Bengio. Deep learning, volume 1. MIT Press, 2016
> Junying Li, Zichen Yang, Haifeng Liu, and Deng Cai. Deep rotation equivariant network. Neurocomputing, 290:26–33, 2018

---

> > ### Author Response · Authors · 2019-11-06
> > **First thoughts [Continuation]**
> >
> > Q: In equation 8 (second equality), you have said f_R^l(F^l) = A(f_R^l(F^l)). How can this be true if A is not the identity? Giving the benefit of the doubt, this could just be a typo.
> > A: Note that both R's are different in equations 8 and subsequent. $\mathcal{R}$ refers to a co-attentive equiv. mapping, while $R$ refers to a conventional equiv. mapping.
> >
> > Q: Please define \odot (I think it’s element-wise multiplication)
> > A: We will do it in a subsequent version of the paper.
> >
> > Q: Are you using row-vector convention? That would resolve some of my confusion with the maths.
> > A: Please see 1. in our previous comment.
> >
> > Q: You define the matrix A as in the space [0,1]^{n x m}. While sort of true, it is more precise to note that each column is actually restricted to a simplex, so A lives in a subspace of [0,1]^{n x m}.
> > A: You are completely right. We will do it in a subsequent version of the paper.
> >
> > Q:  I think it would have been easier just say that you are using a roto-translation or p4m equivariant CNN with attention after each convolution. Then you could derive the constraint on the attention matrices to maintain equivariance. It would be easier to follow and make easy connections with existing literature on the topic.
> > A: Our approach is more general and it is actually extendable to any group producing cyclic permutations as a result of the equivariant mapping. We believe that your way of defining the paper flow is very good and we actually intended to to that in our paper as well: starting from an (1) equiv. net in general, then (2) stating that we use attention on top of it, (3) imposing rules/modifications on the self-attention mechanism used and, finally, (4) adding constraints to the parameters learned in the attention mechanism block.
> > We hope that after implementing the reviewers observations and utilizing simpler notation, the paper flow will be much clearer and easier to follow and it will be easier to connect to existing literature in the topic.
> >
> > Once again, thank you very much for your time, attention and extremely useful commentaries. Please let us know if you have any further questions or comments. We are happy to respond them all :)
> >
> > Best regards,
> > The Authors.

---

### Official Review · AnonReviewer2 · 2019-10-23
**Official Blind Review #2**

**Rating:** 8

**Review:**

[Update after rebuttal period]

While I still find the paper somewhat hard to parse, the revision and responses have addressed most of my concerns. I think this paper should be accepted, because it presents a novel and non-trivial concept (rotation-equivariant self attention).


[Original review]

The authors propose a self-attention mechanism for rotation-equivariant neural nets. They show that introduction of this attention mechanisms improves classification performance over regular rotation-equivariant nets on a fully rotational dataset (rotated MNIST) and a regular non-rotational dataset (CIFAR-10).

Strengths:
+ States a clear hypothesis that is well motivated by Figs. 1 & 2
+ Appears to accomplish what it claims as contributions
+ Demonstrates a rotation-equivariant attention mechanism
+ Shows that its introduction improves performance on some tasks

Weaknesses:
- Unclear how the proposed attention mechanism accomplishes the goal outlined in Fig. 2d
- Performance of the authors' evaluations of the baselines is lower than reported in the original papers, casting some doubt on the performance evaluation
- The notation is somewhat confusing and cumbersome, making it hard to understand what exactly the authors are doing
- No visualisation or insights into the attention mechanism are provided

There are three main issues detailed below that I'd like to see addressed in the authors' response and/or a revised version of the paper. If the authors can address these concerns, I am willing to increase my score.

1. The motivation for the attention mechanism (as discussed in the introduction and illustrated in Fig. 2) seems to be to find patterns of features which commonly get activated together (or often co-occur in the training set). However, according to Eq. (9), attention is applied separately to orientations of the same feature ($A_i$ is indexed by i, the channel dimension), and not across different features. Since the attention is applied at each spatial location separately, such mechanism only allows to detect patterns of relative orientations of the same feature appearing at the same spatial location. The motivation and utility of such formulation is unclear, as it appears to be unable to solve the toy problem laid out in Fig. 2. Please clarify how the proposed mechanism would solve the toy example in Fig. 2.

2. The only real argument that the proposed mechanism is useful are the numbers in Table 1. However, the experimental results for CIFAR-10 are hard to compare to the baselines because of differences in reported and reproduced results. I would appreciate a clarification about the code used (was it published by the authors of other papers?) and discussion of why the relative improvement achieved by the proposed method is not an artefact of implementation or optimisation issues.

3. The exposition and notation in section 3.1 is very hard to follow and requires substantial improvement. For instance, the sections "Attention and self attention" and "Compact local self attention" seem to abstract from the specific case and use x and y, but it is unclear to me what x and y map to specifically. Maybe also provide a visualization of how exactly attention is applied.


Minor comments/questions:

- If the attention is applied over the orientations of the same feature, why does it improve the performance on Rotated MNIST (which is rotation invariant)?

- I assume the attention matrix $A_i$ is different for each layer, because the features in different layers are different and require different attention mechanisms. However, unlike F and K, A is not indexed by layer l.

- It would be good to provide the standard deviation for the reported results on CIFAR-10 to see if the improvement is significant.

**Experience Assessment:**

I have published one or two papers in this area.

**Review Assessment: Checking Correctness Of Derivations And Theory:**

I assessed the sensibility of the derivations and theory.

**Review Assessment: Checking Correctness Of Experiments:**

I assessed the sensibility of the experiments.

**Review Assessment: Thoroughness In Paper Reading:**

I read the paper thoroughly.

---

> ### Author Response · Authors · 2019-11-06
> **First thoughts**
>
> Dear reviewer 2,
>
> First of all thank you very much for thorough review and, of course, for your time. Thank you very much for supporting our work as well.
>
> 1. Regarding *weaknesses*
>
> Q: Unclear how the proposed attention mechanism accomplishes the goal outlined in Fig. 2d.
> A: We will make it clearer in our following version of the paper.
>
> Q: The notation is somewhat confusing and cumbersome, making it hard to understand what exactly the authors are doing.
> A: We consistently obtained from all the reviewers the observation that the notation and technical jargon utilized to explain our approach was excessive. We agree with this.
>
> Our actual motivation to utilize the current notation comes from the  *Default Notation* section of the conference template's ( https://github.com/ICLR/Master-Template/blob/master/archive/iclr2020.zip ). Here, it is encouraged to use the notation utilized in Godfellow et. al. (2016), which (to some extent) contributes to the excessive technical jargon. Several of the equations appearing in our *Section 2* are indeed slight modifications of equations appearing in the *Convolutional Networks* chapter of Godfellow et. al. (2016), which we subsequently utilize to define our own approach.
>
> Furthermore, we accept that we did not introduce relevant terms properly. This is also related to the fact that we understood the provided notation as (intended to be) standard for all submissions and, hence, we did not feel obliged to introduce notation defined in there (e.g. row-vector convention). Naturally, this also contributes to the non-clarity of the derivations.
>
> That said, we accept that the aforementioned facts are not the only reasons contributing to non-clarity. We will work hard at making them much clearer in a subsequent version of our work. We will utilize a simpler, well-introduced notation as well.
>
> Q: No visualisation or insights into the attention mechanism are provided
> A: We are doing our best to provide visualizations that faithfully compare conventional equiv. mappings and co-attentive equiv. mappings. We hope to incorporate them in a subsequent version of the paper.
>
> 2. Regarding *Main issues*:
>
> 2.2: The only real argument that the proposed mechanism is useful are the numbers in Table 1. However, the experimental results for CIFAR-10 are hard to compare to the baselines because of differences in reported and reproduced results. I would appreciate a clarification about the code usJunying Li, Zichen Yang, Haifeng Liu, and Deng Cai. Deep rotation equivariant network. Neurocomputing, 290:26–33, 2018ed (was it published by the authors of other papers?) and discussion of why the relative improvement achieved by the proposed method is not an artefact of implementation or optimization issues.
> A: Our implementation is based on the implementations found online for each of the baselines. For DREN (Li et. al., 2018) we utilize the code provided for Caffe (rot-MNIST) and Tensorflow (CIFAR) (https://github.com/ZJULearning/DREN). The results reported in Table 1 are based on these implementations. We utilize their training strategies as faithfully as possible.
>
> For G-CNNs (Cohen and Welling, 2016) we re-implemented those provided in https://github.com/tscohen/gconv_experiments/tree/master/gconv_experiments, since we needed to perform code multiple updates, as it is written in a now obsolete version of chainer. We do utilize GrouPy (https://github.com/tscohen/GrouPy ) with PyTorch support in our experiments (provided by the authors and other contributors).
>
> Our  reported results emerge from this implementation. We did contact the authors for further information about the experiments with ResNet44. As of now, we are running further experiments based on additional information kindly provided by them and hope to be able to update these results in our the following version of our work. Note that our "implementation problem" boils down to the vanilla ResNet44 and therefore has nothing to do with their proposed method.
>
> Regarding DREN nets, Li et. al., (2018) do not report results in CIFAR-10 for fully equiv. networks. This is due to the fact that the performance strongly degrades when adding more isotonic layers (see Li et. al. (2018) for further details). In our experiments we modify their training setting, to allow the same hyperparameter settings to be used for all methods (i.e., vanilla NIN, r_x4, a-r_x4) and so, provide a reliable comparison. This is why the reported values strongly differ in these cases (please see *Training convergence of equivariant networks* in our work for further details).
>
> Ian Goodfellow, Yoshua Bengio, Aaron Courville, and Yoshua Bengio. Deep learning, volume 1. MIT Press, 2016
>
> Junying Li, Zichen Yang, Haifeng Liu, and Deng Cai. Deep rotation equivariant network. Neurocomputing, 290:26–33, 2018

---

> > ### Author Response · Authors · 2019-11-06
> > **First thoughts [Continuation]**
> >
> > 2.3: The exposition and notation in section 3.1 is very hard to follow and requires substantial improvement. For instance, the sections "Attention and self attention" and "Compact local self attention" seem to abstract from the specific case and use x and y, but it is unclear to me what x and y map to specifically. Maybe also provide a visualization of how exactly attention is applied.
> > A: We will provide a clearer derivation with simpler notation in a subsequent version of our work. A visualization of the method is a good idea as well.
> >
> > 3. Regarding *Minor comments/questions*:
> >
> > Q: If the attention is applied over the orientations of the same feature, why does it improve the performance on Rotated MNIST (which is rotation invariant)?
> > A: Interesting question. Note that invariance is achieved in the last layer of an equiv. net. As a result, all of the internal mappings up to the last one are equivariant, as opposed to invariant. Furthermore, although numbers in the dataset indeed appear at random positions, it does not imply that all possible orientations of a learned filter must contribute equally for all possible input features (consider for example a dataset composed of the 4 rotated versions of the face shown in Fig. 2a). Therefore (1) it is reasonable to expect improvements if we focus on those feature orientations that actually compose a given pattern, and, (2) it is mandatory not to disrupt equivariance on the course of the attention procedure.
> >
> > Q: I assume the attention matrix $\boldsymbol{A}_{i}$ is different for each layer, because the features in different layers are different and require different attention mechanisms. However, unlike F and K, A is not indexed by layer l.
> > A: It is indeed a typo. Sorry about that.
> >
> > Q: It would be good to provide the standard deviation for the reported results on CIFAR-10 to see if the improvement is significant.
> > A: I agree that it would be a good practice. However, note that we utilized the same evaluation measurements as the corresponding baselines (e.g. we provide std.dev. in rot-MNIST when using G-Convs, just as Cohen and Welling, (2016) did). To provide a fair comparison in CIFAR-10, we would need to re-do multiple experiments of the baselines as well. If time suffices, we will  perform these experiments as well.
> >
> > Once again, thank you very much for your time, attention and extremely useful commentaries. Please let us know if you have any further questions or comments. We are happy to respond them all :)
> >
> > Best regards,
> > The Authors.

---

### Official Review · AnonReviewer1 · 2019-10-25
**Official Blind Review #1**

**Rating:** 6

**Review:**

This paper describes an approach to applying attention in equivariant image classification CNNs so that the same transformation (rotation+mirroring) is selected for each kernel. For example, if the image is of an upright face, the upright eyes will be selected along with the upright nose, as opposed to allowing the rotation of each to be independent. Applying this approach to several different models on rotated MNIST and CIFAR-10 lead to smaller test errors in all cases.

Overall, this is a good idea that appears to be well implemented and well evaluated.  It includes an extensive and detailed bibliography of relevant work. The approach seems to be widely applicable. It could be applied to any deep learning-based image classification system. It can be applied to additional transformations beyond rotation and mirroring.

The one shortcoming of the paper is that it takes a simple idea and makes it somewhat difficult to follow through cumbersome notation and over-mathmaticization. The ideas presented would be much clearer as an algorithm or more code-like representation as opposed to as equations. Even verbal descriptions could suffice. The paper is also relatively long, going onto the 10th page. In order to save space, some of the mathematical exposition can be condensed.

In addition, as another issue with clarity, the algorithm has one main additional hyperparameter, r_max, but the description of the experiments does not appear to mention the value of this hyperparameter. It also states that the rotated MNIST dataset is rotated on the entire circle, but not how many fractions of the circle are allowed, which is equivalent to r_max.

**Experience Assessment:**

I have read many papers in this area.

**Review Assessment: Checking Correctness Of Derivations And Theory:**

I assessed the sensibility of the derivations and theory.

**Review Assessment: Checking Correctness Of Experiments:**

I assessed the sensibility of the experiments.

**Review Assessment: Thoroughness In Paper Reading:**

I read the paper at least twice and used my best judgement in assessing the paper.

---

> ### Author Response · Authors · 2019-11-06
> **First thoughts**
>
> Dear reviewer 1,
>
> First of all thank you very much for thorough review and, of course, for your time. Thank you very much for supporting our work as well.
>
> 1. *Regarding your observations*:
>
> 1.1: "The one shortcoming of the paper is that it takes a simple idea and makes it somewhat difficult to follow through cumbersome notation and over-mathmaticization. The ideas presented would be much clearer as an algorithm or more code-like representation as opposed to as equations."
> A: We consistently obtained from all the reviewers the observation that the notation and technical jargon utilized to explain our approach was excessive. We agree with this.
>
> Our actual motivation to utilize the current notation comes from the  *Default Notation* section of the conference template's ( https://github.com/ICLR/Master-Template/blob/master/archive/iclr2020.zip ). Here, it is encouraged to use the notation utilized in Godfellow et. al. (2016), which (to some extent) contributes to the excessive technical jargon. Several of the equations appearing in our *Section 2* are indeed slight modifications of equations appearing in the *Convolutional Networks* chapter of Godfellow et. al. (2016), which we subsequently utilize to define our own approach.
>
> Furthermore, we accept that we did not introduce relevant terms properly. This is also related to the fact that we understood the provided notation as (intended to be) standard for all submissions and, hence, we did not feel obliged to introduce notation defined in there (e.g. row-vector convention). Naturally, this also contributes to the non-clarity of the derivations.
>
> That said, we accept that the aforementioned facts are not the only reasons contributing to non-clarity. We will work hard at making them much clearer in a subsequent version of our work. We will utilize a simpler, well-introduced notation as well.
>
> 1.2: "The paper is also relatively long, going onto the 10th page. In order to save space, some of the mathematical exposition can be condensed."
> A: We will consider this in our camera ready version as well.
>
> 1.3: "as another issue with clarity, the algorithm has one main additional hyperparameter, r_max, but the description of the experiments does not appear to mention the value of this hyperparameter"
> A: Indeed r_max is an additional hyperparameter of rotation equivariant networks. However, as we perform attention on the resulting maps from multiple baseline rotation equivariant networks (i.e. G-CNNs (Cohen and Welling, 2016) and DREN nets (Li et. al., 2018)) we are not able to select a different r_max as that already implemented in the baselines. Therefore, although it is indeed a hyperparameter, it is defined in beforehand by the corresponding baselines.
>
> We will introduce a clear definition of the utilized baselines, such that this becomes precise and clear. Thank you for pointing this out.
>
> 1.4: "It also states that the rotated MNIST dataset is rotated on the entire circle, but not how many fractions of the circle are allowed, which is equivalent to r_max"
> A: This is very related to the last question. This will be addressed by following the previous statement.
>
> Once again, thank you very much for your time, attention and extremely useful commentaries. Please let us know if you have any further questions or comments. We are happy to respond them all :)
>
> Best regards,
> The Authors.
>
> Taco Cohen and Max Welling. Group equivariant convolutional networks. In International conference on machine learning, pp. 2990–2999, 2016.
> Ian Goodfellow, Yoshua Bengio, Aaron Courville, and Yoshua Bengio. Deep learning, volume 1. MIT Press, 2016
> Junying Li, Zichen Yang, Haifeng Liu, and Deng Cai. Deep rotation equivariant network. Neurocomputing, 290:26–33, 2018

---

### Author Response · Authors · 2019-11-09
**On the updated revision**

Dear reviewers,

We have submitted a new revision of our work. In this comment, we summarize the performed changes.

* Mayor Changes *
---------------------------
- We modified the notation utilized across the document such that: (1) it is easier to read and (2) easier to connect to existing literature in the topic ( Based on Rev. 1, 2, 3)
- We reduced the extension of the paper (Based on Rev. 1)
- Substantial changes in the *Learning equivariant neural networks* subsection (Based on Rev. 3)

* Other Changes *
--------------------------
- Clearer definitions provided (Based on Rev. 2, 3)

* Questions to the reviewers*
------------------------------------------
1. Regarding the visualization of the model:
Reviewer 2 correctly signalized that we lack providing insights into the model. Unfortunately, we are having a hard time finding a good way to do so and we would like to ask the reviewers for further guidance.

Possibilities we have thought of:
- We could show the effect of rotating the input and analyzing differences in the output. However, since it is proven that equivariance holds, this provides no further insights into the network.
- We could show the effect of the attention mechanism in the model, i.e. maps before attention and after attention. However, since the model has been optimized to utilize softmax in the process, we believe that showing the effects of the softmax on feature maps might provide few insight as well.
- As co-attentive and conventional networks are solving two different optimization problems, we find it very difficult to provide a fair comparison between learned feature representations (since they are not restricted to be similar).

We would sincerely appreciate any further thoughts or comments on this issue.

2. Regarding the performance of the model:
Reviewer 2 signalized his concern about the obtained results. In order to evaluate the real contribution of our method, we have performed further experiments with hyperparameter optimization. By doing so, we have obtained better results on a-P4CNN (around 1.85% test error). We believe that by performing hyperparameter optimization for the remaining models we will boost the performance measurements obtained so far.
Our question is: Are these experiments a good way to address these concerns? If so, we will continue with our experiments and update our results in a subsequent submission. If this is not the case, we would like to ask you for ways to address them properly.

* Expected additions in the future *
--------------------------------------------------
- We are working on proving that a mapping is equivariant to cyclic permutations iff it possesses the form of a circulant matrix.
- Visualization of the method itself will be provided
- Hopefully visualizations and comparisons on the learned feature representations.

*Important*
------------------
If we missed something you find important in our update. Please let us know.


In advance, thank you very much for your time. We are looking forward to hearing from you.

Best regards,
The authors.

---

> ### Comment · AnonReviewer2 · 2019-11-09
> **Most crucial point not addressed**
>
> Please try to address my main concern, ideally during the discussion period in case I am misunderstanding something:
>
> 1. The motivation for the attention mechanism (as discussed in the introduction and illustrated in Fig. 2) seems to be to find patterns of features which commonly get activated together (or often co-occur in the training set). However, according to Eq. (9), attention is applied separately to orientations of the same feature ( is indexed by i, the channel dimension), and not across different features. [...] The motivation and utility of such formulation is unclear, as it appears to be unable to solve the toy problem laid out in Fig. 2. Please clarify how the proposed mechanism would solve the toy example in Fig. 2.

---

> > ### Author Response · Authors · 2019-11-10
> > **Response to most crucial point**
> >
> > Dear Reviewer 2,
> >
> > Thank you very much for your fast response.
> >
> > Excuse us for not responding your question yet. Our intention was to first modify the notation in the paper itself so that it is easier to address your question.
> >
> > Q: The motivation for the attention mechanism (as discussed in the introduction and illustrated in Fig. 2) seems to be to find patterns of features which commonly get activated together (or often co-occur in the training set). However, according to Eq. (9), attention is applied separately to orientations of the same feature ( is indexed by i, the channel dimension), and not across different features. Since the attention is applied at each spatial location separately, such mechanism only allows to detect patterns of relative orientations of the same feature appearing at the same spatial location. The motivation and utility of such formulation is unclear, as it appears to be unable to solve the toy problem laid out in Fig. 2. Please clarify how the proposed mechanism would solve the toy example in Fig. 2.
> >
> > A: We understand your concern. In fact a direct approach should address the problem illustrated in the introduction and Fig. 2 by applying an attention mechanism that acts simultaneously on $(r, \lambda)$ as you correctly describe. However, we are able to simplify the problem by taking advantage of the equivariance property of the network.
> >
> > Consider $p$, the input to of a roto-translational conv. $f_{R}: Z^{2} \rightarrow \Theta \times Z^{2} \times \Lambda $ as outlined in eq. 3 in the new submission, and $\Theta$ be the set of rotations of 90 degrees. i.e. $\text{dim}(\Theta)=4$ (in general $\Theta$ is also included in the domain of $f_{R}$ but we remove it for simplicity; i.o.w. $f_{R}$ is located at the first layer of a neural network). Now, let $A = \{\{f_{R}(p)(u)\}_{r=1}^{4}\}_{\lambda=1}^{\Lambda}$ be the matrix of dimension $4 \times \Lambda$ at a certain position $u$, consisting of the $4$ oriented responses for each of the  $\Lambda$ learned representations.
> >
> > Due to the fact that the the vectors $\{f_{R}(p)(u)\}_{r=1}^{4}(\lambda)$ permute cyclically as a result of $f_{R}$, it is mandatory, as outlined in the paper, to ensure equivariance to cyclic permutations in each $A(\lambda)$. At first sight, one might think that there's no connection between multiple $\lambda$'s in $A$, and, therefore, in order to exploit co-occurences, one must impose additional constraints along the $\lambda$ axis. Extremely interestingly, however, is the fact that not just one mapping but the entire layer (i.e. all $\Lambda$ mappings) behaves predictively as a result of equivariance and, hence, there is already an implicit restriction between mappings along $\lambda$.
> >
> > Consider, for instance, the input $\theta_{i} p$, a $\theta_{i}$-rotated version of $p$. By virtue of the equivariance property of $f_{R}$, we have (locally) that $f_{R}(\theta_{i} p) = \mathcal{P}^{i}(f_{R}(p))$ and thus $f_{R}(\theta_{i} p)(u,r,\lambda)= \mathcal{P}^{i}(f_{R}(p)(u, r, \lambda)) \forall \lambda \in \Lambda$. In other words, the equivariance property of the mapping assures that all of the $\Lambda$ feature maps move "synchronously" as a function of a rotation in the input. Resultantly, there is already an implicit constraint regarding how $A$ behaves along the $\lambda$ axis. Note that if we have an equivariant attention mechanism $\mathcal{A}$, $\mathcal{A}(f_{R}(\theta_{i} p))(u,r,\lambda)= \mathcal{P}^{i}(\mathcal{A}(f_{R}(p))(u, r, \lambda)) \forall \lambda \in \Lambda$ must hold as well. As a result, all of the $\Lambda$ attention mechanisms applied along $r$ must move "synchronously" as a function of a rotation in the input.
> >
> > As a matter of fact and due to computational reasons, we utilize in our implementation a matrix with the same form of $A$ to store the coefficients of our attention mechanism (since each $\tilde{A}$ in $\mathcal{A_{\lambda}^{C}}$ is actually fully defined by a vector). Very interesting is it to see that if one rotates the input of a layer, all the $\Lambda$ instances $\mathcal{A_{\lambda}^{C}}$ beautifully synchronously rotate their "attention mask" accordingly.
> >
> > We hope that our answer sheds a bit more light into the behavior of our algorithm.
> >
> > If you have any comment about this or the other questions stated above, we are happy to hear them :)
> >
> > Best regards,
> > The authors.

---

> > > ### Comment · AnonReviewer2 · 2019-11-11
> > > **Can you illustrate it?**
> > >
> > > Thanks for the quick response. Unfortunately I cannot wrap my head around your explanation. I even discussed it with a colleague and we couldn't figure it out together.
> > >
> > > Could you try to illustrate, using the toy example from Fig. 2, how attention acts and what values the attention matrices A would assume in the case of this toy example? In other words, illustrate graphically (a) what is the dimensionality of the attention tensor in a given layer, (b) how are weights shared due to cyclical constraints, (c) based on what tensor of (d) what dimensionality is it computed, and (e) how does it manipulate the activations.
> > >
> > > It seems like a diagram with boxes and (annotated) arrows would do the job way better than pages of math.

---

> > > > ### Author Response · Authors · 2019-11-12
> > > > **Illustrated explanation**
> > > >
> > > > Dear reviewer 2,
> > > >
> > > > Here's a link to a graphical explanation of the attention mechanism:
> > > > https://www.dropbox.com/s/h9dbzija8ml2646/attention_graphical.png?dl=0
> > > > There are 3 matrices of 4x4 attention weights one for each of eye, mouth, eyes.
> > > >
> > > > *Connections to our previous response:*
> > > > The equivariance property of the mapping assures that all of the $\Lambda$ feature maps (here mouth, nose, eyes) move "synchronously" as a function of a rotation in the input. Resultantly, there is already an implicit constraint regarding how $f_{R}$ behaves along the $\lambda$  axis. Consequently, if one rotates the input of a layer, all the $\Lambda$ attention instances beautifully synchronously rotate their "attention mask" accordingly.
> > > >
> > > > *Extension on (b):*
> > > > We will add a section in the appendix showing that a linear mapping of the form $xA$ is equivariant to cyclic permutations iff $A$ is a circulant matrix.
> > > >
> > > > **EDIT:** We found that it has already been proven that a matrix $A$ is circulant if and only if it is equivariant with respect to cyclic shifts of the domain and hence it is unnecessary to introduce a self-developed proof. We will introduce the corresponding references in our derivations.
> > > >
> > > >
> > > > We hope that our answer sheds more light into the functionality of our algorithm.
> > > >
> > > > If you have any comment about this or the other questions stated above, we are happy to hear them :)
> > > >
> > > > Best regards,
> > > > The authors.

---

### Author Response · Authors · 2019-11-15
**Summary of changes during the rebuttal period**

Dear reviewers,

Once again, thank you very much for your helpful and thorough reviews, which resulted in a largely improved version of our work.

In this comment we provide a brief summary of the changes done to our work during this rebuttal time.

*Changes:*
----------------
- We modified the notation utilized across the document such that: (1) it is easier to read and (2) easier to connect to existing literature in the topic.
- We included a visual representation of how our algorithm works.
- We modified definitions and restructured the paper such that its readability and congruence was largely improved.
- Based on the discussions with Reviewer 2, we included in the appendix an additional section with a meticulous discussion about how co-occurrent attention is obtained via the proposed framework.
- Additional relevant references were incorporated.

Thank you very much for your time and your attention.

Best regards,
The authors.

---

### Public Comment · ~Gladis_Ne_Limes1 · 2023-09-22
**re**

I recommend read this blog about software development in warehousing https://mlsdev.com/blog/technology-in-warehousing

---

### Decision · Program_Chairs · 2019-12-19

**Decision:**

Accept (Poster)

**Comment:**

The paper proposes an attention mechanism for equivariant neural networks towards the goal of attending to co-occurring features. It instantiates the approach with rotation and reflection transformations, and reports results on rotated MNIST and CIFAR-10. All reviewers have found the idea of using self-attention on top of equivariant feature maps technically novel and sound. There were some concerns about readability which the authors should try to address in the final version.